# Study of Polyvinyl Alcohol-SiO_2_ Nanoparticles Polymeric Membrane in Wastewater Treatment Containing Zinc Ions

**DOI:** 10.3390/polym13111875

**Published:** 2021-06-04

**Authors:** Simona Căprărescu, Cristina Modrogan, Violeta Purcar, Annette Madelene Dăncilă, Oanamari Daniela Orbuleț

**Affiliations:** 1Inorganic Chemistry, Physical Chemistry and Electrochemistry Department, Faculty of Applied Chemistry and Materials Science, University POLITEHNICA of Bucharest, 1-7 Polizu Str., 011061 Bucharest, Romania; simona.caprarescu@upb.ro; 2Analytical Chemistry and Environmental Engineering Department, Faculty of Applied Chemistry and Materials Science, University POLITEHNICA of Bucharest, 1-7 Polizu Str., 011061 Bucharest, Romania; madelene.dancila@upb.ro (A.M.D.); oanamari.orbulet@upb.ro (O.D.O.); 3National Institute for Research & Development in Chemistry and Petrochemistry—ICECHIM, Splaiul Independentei No. 202, 060021 Bucharest, Romania

**Keywords:** silica nanoparticles, electrodialysis, wastewater, zinc ions, ionic conductivity

## Abstract

The main goal of the present paper was to synthesize the polyvinyl alcohol-SiO_2_ nanoparticles polymeric membrane by wet-phase inversion method. The efficiency of prepared membranes (without and with SiO_2_) was investigated using a versatile laboratory electrodialysis system filled with simulated wastewaters that contain zinc ions. All experiments were performed at following conditions: the applied voltage at electrodes of 5, 10 and 15 V, a concentration of zinc ions solution of 2 g L^−1^, time for each test of 1 h and at room temperature. The demineralization rate, extraction percentage of zinc ions, current efficiency and energy consumption were determined. The polymeric membranes were characterized by Fourier Transforms Infrared Spectroscopy-Attenuated Total Reflection (FTIR-ATR), Scanning Electron Microscopy (SEM) and Electrochemical Impedance Spectroscopy (EIS). The higher value of percentage removal of zinc ions (over 65%) was obtained for the polymeric membrane with SiO_2_ nanoparticles, at 15 V. The FTIR-ATR spectra show a characteristic peak located at ~1078 cm^−1^ assigned to the Si-O-Si asymmetrical stretching. SEM images of the polymeric membrane with SiO_2_ nanoparticles show that the nanoparticles and polymer matrix were well compatible. The impedance results indicated that the SiO_2_ nanoparticles induced the higher proton conductivity. The final polymeric membranes can be used for the removal of various metallic ions, dyes, organic or inorganic colloids, bacteria or other microorganisms from different natural waters and wastewaters.

## 1. Introduction

The pollution of air, water and soil with metallic ions has attracted great attention of environmentalists and researchers. The most metallic ions (e.g., Zn^2+^, Cu^2+^, Fe^2+^, Cd^2+^, Pb^2+^, Ni^2+^, Hg^2+^) released into the environment from industrial activities (e.g., electroplating, smelting, battery manufacture, tanneries, paint and pigments manufacture, pesticides, printing and photographic industries, mining and refining processes) [1,2], can have negative effects on plants, animals and human health, because are toxic, non-biodegradable, persistent, have potential to be bio-accumulated and cannot be metabolized or decomposed [2,3]. The major sources of zinc ions into the environment by wastewaters are electroplating, smelting, alloy manufacturing, ceramics, textiles, fertilizers, pigments, batteries and mining industry effluents [2,3]. The European Environment Protection Agency and the World Health Organization established lists with the dangerous substances and inorganic pollutants where zinc has been included [2,4,5]. Based on the lists, the World Health Organization, Environment Protection Agency, European Ministry of Environment and Forest, established limits of discharge of zinc from industrial activities into the environment as follows: for estuaries and marine waters of 40 µg L^−1^, for freshwater between 45 and 500 µg L^−1^ and for wastewater of 5 mg L^−1^ [2,3,4]. Zinc is considered an essential element for the aquatic ecosystems, natural flora and fauna and for human health. It was reported [2,5] that the concentration of 120 µM of zinc may cause negative and irreversible effects for plants and animals (e.g., micronucleus induction in plant *Vicia faba* [6], kill some organisms, invertebrates and even vertebrate fish [2,6]. The World Health Organization and The Institute of Medicine recommended as a tolerable level of zinc from foods of 15 mg day^−1^ for adults [7,8]. The tolerable level of zinc from supplements was established for an adult of 40 mg day^−1^ [8] and for children of 2 mg day^−1^ [9]. It was reported that over tolerable limit of zinc can produce gastrointestinal disturbance, i.e., irritability, loss of appetite, muscular stiffness, diarrhea [5,10]. The persons that work in different industries (e.g., galvanizing and in manufacturing brass another alloys, batteries and pigments) and are exposure to zinc at levels between 100 and 500 mg day^−1^ [2,10], can have health problems, such as: lethargy, hyperamylasemia, pancreatitis, pulmonary edema, renal insufficiency, and neurological disturbances [2,10,11].

Several conventional methods have been used for the treatment and removal of heavy metals from water and wastewater, such as: chemical precipitation, coagulation, flotation, ion exchange, solvent extraction, and chemical adsorption [3,12,13]. These conventional methods have limitations, such as: large amount of sludge, extra operational cost for sludge disposal, large amount of chemicals for removal of metals from waters and wastewaters, involves large activation energy, low selectivity [11,12,13]. Membrane technologies [14,15,16], such as: electrodialysis, ultrafiltration, reverse osmosis, nanofiltration, has been successfully applied in the last decades, at the laboratory scale and at the industrial level, due to the major advantages such as: high efficiency, can function at low temperature, does not require additional chemicals, low environmental impact [14,16,17,18].

Electrodialysis is a membrane separation method for the removal of ionized species from water and/or wastewater by transporting ions through an ion exchange membrane, semi-permeable membrane or ion-permeable membrane to other solution under applying an electric potential directed perpendicular to the membrane plane. The membranes are positioned between the cell compartments of the electrodialysis. When electric current is passed through the cell compartments of the electrodialysis, anions migrate to the anode electrode and are retained by the negatively charged membranes, while the cations migrate to the cathode electrode and are retained by the positively charged membranes. At the final of the electrodialysis process with three compartments, as a result of the transport process, the solution from the central compartment will be diluted in pollutants, and the solution from the anodic compartment will be enriched/concentrated in pollutants [14,17,18,19]. Electrodialysis has been successfully applied at the laboratory scale and at the industrial level for treated waters and wastewaters containing contaminants and products, such as: heavy metal ions [14,17,18], acids [19], dyes [16,20], and organic matter [16,19]. The wastewaters may come from different industries, such as: chemistry [18,19,20], (bio)chemistry [21], food processing [22], pharmaceutical [18,21], mining and refining [15,17,22,23]. Electrodialysis was successfully applied for the removal of contaminants from different waters and wastewaters due to their advantages, such as: high separation selectivity and efficiency, does not require specialized equipment, operates without noise, low maintenance, low capital cost, low space and material requirements [14,17,20,21,22].

In the last years, researchers have synthesized polymeric membranes that contains different types of natural (e.g., cellulose and cellulose derivates, chitin and chitosan [23,24]) or synthetic polymers (e.g., polyvinyl alcohol, polyamide, polystyrene, polysulfone, polyethersulfone [25,26,27]) and metal oxide nanoparticles (e.g., silica (SiO_2_), zinc oxide (ZnO), titanium dioxide (TiO_2_), zirconium (ZrO_2_), iron oxide (Fe_2_O_3_) [28,29,30]). Pereira et al. [30] synthetized the biohybrid membranes by association of electrospun hydrolysis–resistant polyvinyl alcohol membranes and free-living bacteria for the removal of pollutants. The studies indicated that than 46% of the hexavalent chromium and the phenol content were removed from a tannery effluent. Torasso et al. [31] developed electrospun membranes containing iron oxide nanoparticles inside polyvinyl alcohol nanofibers for arsenic adsorption. They demonstrated that the prepared membrane has an enhanced arsenic adsorption capacity (52 mg g^−1^). SiO_2_ nanoparticles have used in the synthesis of different membranes and applied in the membrane processes due to their special properties, such as strong surf ace energy, small size, thermal resistance, fine suspension in aqueous solution and relatively inert from an ecological point of view [30,32,33]. The polymeric membranes with SiO_2_ nanoparticles were successfully applied for water and wastewater treatment due to their advantages, such as: exhibit excellent permeability, higher selectivity, chemical stability, mechanical and thermal resistance, high hydrophilicity, fouling resistance, assure higher fluxes for water [29,31,32,33]. Rosdi et al. [33] prepared a chitosan/silica composite membrane for separate of lead(II) ion from aqueous solution. They related that the adsorption efficiency was 89.27% for composite membrane as compared to 11.50% of pure chitosan membrane, at optimum pH of 7.0 and an initial concentration of 220 mg L^−1^. In our previously studies, we have successfully applied a mini-electrodialysis system and different membranes containing ion exchange resins for the treatment of a synthetic effluent which contained zinc ions. The obtained results indicated that the removal ratio of zinc ions was higher (over 75%) when higher concentrated solution was treated (6 g L^−1^), at a constant applied voltage of 5 V, after 1.5 h of treatment [34].

In this work, we prepared and investigated the performance of polymeric membrane without and with SiO_2_ nanoparticles for zinc ions (Zn^2+^) removal from a simulated wastewater. The prepared polymeric membranes (without and with SiO_2_) were tested at different conditions using a new versatile laboratory electrodialysis system, own construction. The performance of the laboratory electrodialysis system was evaluated in terms of demineralization rate, extraction percentage of zinc ions, current efficiency and energy consumption. The prepared polymeric membranes were characterized by Fourier Transforms Infrared Spectroscopy-Attenuated Total Reflection (FTIR-ATR), Scanning Electron Microscopy (SEM) and Electrochemical Impedance Spectroscopy (EIS).

## 2. Materials and Methods

### 2.1. Materials and Reagents

Silicon dioxide nanopowder (SiO_2_, between 10 and 20 nm size) was purchased from Sigma–Aldrich (Merck, Redox Lab Supplies Com SRL, Bucharest, Romania).

Acrylonitrile, vinyl acetate and dimethyl sulfoxide were supplied by Sigma-Aldrich (Merck KGaA, Darmstadt, Germany) and used without further purification.

### 2.2. Synthesis of Polyvinyl Alcohol-SiO_2_ Nanoparticles Polymeric Membrane

Polyvinyl alcohol solution, due to stabilization effect, was used for the preparation of polymeric membranes. This solution was prepared in the laboratory by following procedure: firstly, a solution of polyvinyl acetate was obtained by radical polymerization of vinyl acetate in methanol (80:20, wt.%), at 60 °C, under refluxing, using as initiator the 2,2′-Azobis(2-methylproprionitrile) (98%, Sigma-Aldrich, Merck KGaA, Darmstadt, Germany). After that, to obtain the final solution of polyvinyl alcohol, a mixture of polyvinyl acetate and methanol (40%) was catalyzed in the presence of 2% of sodium hydroxide (NaOH) (Merck, Redox Lab Supplies Com SRL, Bucharest, Romania) (calculated to amount of polyvinyl acetate), at 50 °C. Vinyl acetate and methanol were supplied by Sigma-Aldrich (Merck, Redox Lab Supplies Com SRL, Bucharest, Romania).

The polyvinyl alcohol-SiO_2_ nanoparticles polymeric membrane was obtained by wet-phase inversion method, at room temperature (24 ± 1 °C), by following procedure: in a beaker was added a mixture of 3.5 g of copolymers (acrylonitrile (C_3_H_3_N) (70%) and vinyl acetate (C_4_H_6_O_2_) (30%)), and polyvinyl alcohol solution (20%) was completely dissolved in 50 mL of dimethyl sulfoxide ((CH_3_)_2_SO), under constant and uniformly magnetic stirring (300 rpm), at 100 °C (DLAB MS-H-Pro+, AMEX-lab, Bucharest, Romania), for 5 h. After that, 15% wt. SiO_2_, calculated to the polymer mixture, were gradually added to the solution, under magnetic stirring (200 rpm), at 90 °C (DLAB MS-H-Pro+, AMEX-lab, Bucharest, Romania), for another 4 h. The viscous and homogeneous polymeric solution was left overnight at room temperature (24 ± 1 °C) without stirring to remove the air bubbles. The polymeric solution obtained was cast on a detachable glass plate using a manually film applicator (Multicator 411, Erichsen, Minneapolis, MN, USA). To obtaining a uniform film, thickness of 0.25 mm, the spread casting solution was controlled manually adjusting the height of the applicator. After that, the film cast onto the glass plate was dipped immediately into a glass vessel containing deionized water, at room temperature (24 ± 1 °C), to obtain the polyvinyl alcohol-SiO_2_ nanoparticles polymeric membrane. The schematically preparation of the polymeric membrane with SiO_2_ nanoparticles is indicate in Figure 1.

A similar procedure was employed for the synthesis of the polymeric membrane without SiO_2_ nanoparticles.

All prepared polymeric membranes were washed three times with deionized water before being used to remove zinc ions from a synthetic wastewater.

### 2.3. Laboratory Electrodialysis System

Experiment tests were realized using an new versatile laboratory electrodialysis system, own construction, composed from following components: three compartments (anodic, middle and cathodic) made from poly(methyl methacrylate), two pure graphite electrodes (anode and cathode, 99.9% purity) and two identically prepared polymeric membranes. The electrodes were positioned at the extremities of the laboratory electrodialysis system and the prepared polymeric membranes were placed between the compartments (Figure 2).

The dimensions of the laboratory electrodialysis system (Figure 2) were: external diameter of each compartment of 7.5 cm, inside diameter of each compartment of 6.3 cm, thickness of each compartment without O-rings of 0.765 cm, thickness of each compartment with O-rings (T-06879 YATO, Yato Tools Co., Ltd. Shanghai, China), on both sides of the compartment, of 0.985 cm. Thickness of each electrode was 0.935 cm. The working surface area of each prepared polymeric membrane and each graphite electrode was approximately 32 cm^2^.

All compartments of laboratory electrodialysis system (Figure 2) were filled with the same simulated wastewater prepared from zinc(II) sulfate heptahydrate (ZnSO_4_·7H_2_O) (Chimopar SRL, Bucharest, Romania), sulphuric acid (H_2_SO_4_) (98%, Merck KGaA, Darmstadt, Germany) (molar ratio 1:1) and distilled water to obtain a solution containing 2 g L^−1^ of zinc ions. The total volume of simulated wastewater in all compartments was approximately 95 cm^3^. All experiments were realized at different applied voltages at electrodes of 5, 10 and 15 V, using a power supply (AX-3005D, 0–30 VDC, 0–5 A, AXIOMET, Transfer Multisort Elektronik, Lodz, Poland), without recirculation of simulated wastewater, at room temperature (24 ± 1 °C) and a time for each test of 1 h.

The laboratory electrodialysis system performance was evaluated in terms of (i) demineralization rate (D_r_); (ii) percentage removal (P_r_) of zinc ions; (iii) current efficiency (or transport) yields (I_e_) of zinc ions, and (iv) energy consumption (E_c_).

The solutions, before and after experimental tests, were filtered using the filter paper (Whatman paper, 0.8 µm) and were analyzed using a UV-Vis spectrophotometer (SP-830+, Metertech Inc., Nankang, Taipei, Taiwan), at a wavelength of 330 nm, to determine the concentrations of zinc ions.

Demineralization rate (D_r_) was calculated using the following Equation (1) [20,35]:(1)$$D_{r}=(1-\frac{\lambda_{f}}{\lambda_{i}})\cdot 100$$
where λ_i_ and λ_f_ are the conductivities of the initial wastewater and of the final solutions from middle compartment, after 1 h of treatment (mS cm^−1^).

The conductivity of filtered solutions was measured using a multi-parameter analysis (C1010, Consort bvba, Turnhout, Belgium).

The percentage removal of zinc ions (P_r_) was calculated using the following Equation (2) [14,34,36,37]:(2)$$P_{r}=\frac{C_{i}-C_{f}}{C_{i}}\cdot 100$$
where C_i_ and C_f_ are the initial and the final concentrations of zinc ions (mg L^−1^).

The current efficiency (I_e_) was calculated based on the concentrations of solutions using Equation (3) [14,34]:(3)$$I_{e}=\frac{z\cdot \left(C_{i}-C_{f}\right)\cdot V\cdot F}{N\cdot \bar{I}\cdot t}\cdot 100$$
where I_e_ is current efficiency; z is charge number of zinc ions; V is the total volume of simulated wastewater filled in all compartments (L); F is the Faraday constant (96,486 C mol^−1^); N is the number of membrane pairs; Ī is the average current intensity (A); t is the duration time of treatment (s).

The energy consumption (E_c_) during electrodialysis process was calculated by Equation (4) [14,34,35,37]:(4)$$E_{c}=\frac{U}{V}\cdot \int_{t=0}^{t=t_{f}}I(t)\cdot dt$$
where E_c_ is the energy consumption (kWh m^−3^), I is the current intensity (A), U is the applied voltage (V), t is the initial time (h), t_f_ is the final time (h), and V is the total volume of feed solution from all compartments (m^3^).

All prepared polymeric membranes, before and after using in the laboratory electrodialysis system (Table 1), were characterized by Fourier Transforms Infrared Spectroscopy-Attenuated Total Reflection (FTIR-ATR), Scanning Electron Microscopy (SEM), and Electrochemical Impedance Spectroscopy (EIS).

### 2.4. Characterization of Prepared Polymeric Membranes

#### 2.4.1. Fourier Transforms Infrared Spectroscopy-Attenuated Total Reflection (FTIR-ATR)

Infrared spectroscopy study was performed on FTIR spectrophotometer (Perkin ELMER, Ltd., London, UK) in the attenuated total reflection mode (ATR). The FTIR-ATR spectra were recorded at the 4000–400 cm^−1^ interval, by collecting 16 scans for each point, at a spectral resolution of 4 cm^−1^.

#### 2.4.2. Scanning Electron Microscopy (SEM)

The top-surface and cross-section morphologies of the obtained polymeric membranes, cut into pieces of 0.2 cm × 0.2 cm, covered with a thin gold layer and performed in low vacuum mode, were visualized using Scanning Electron Microscopy (S-2600N, HITACHI, Tokyo, Japan).

#### 2.4.3. Electrochemical Impedance Spectroscopy (EIS)

All electrochemical impedance experiments have been carried using a potentiostat/galvanostat (Parstat CS350 Electrochemical System, Wuhan Corrtest Instruments Corp. Ltd., Wuhan, China), associated with an integrated software (ZSimpWin version 3.5, AMETEK Scientific Instruments, Berwyn, PA, USA). The EIS measurements were performed using an electrochemical cell composed by two symmetrical platinum electrodes (Figure 3a) (surface area of 0.1 cm × 0.1 cm, thickness of 0.55 mm, purity 99%), at room temperature (24 ± 1 °C). The small pieces of each wet prepared polymeric membrane (surface area of 0.2 cm × 0.2 cm), were placed between platinum electrodes (Figure 3b) and connected to the electrochemical system, to obtain the main electrical parameters. The applied frequency range was between 100 kHz and 1 Hz, and the applied a.c. amplitude perturbation was 10 mV.

Proton conductivity (σ) of the polymeric membranes (without and with SiO_2_ nanoparticles) was calculated by using the Equation (5) [14,38]:(5)$$\sigma =\frac{l}{R\cdot A}$$
where σ is the proton conductivity (S cm^−1^), l is the membrane thickness (cm), A is the area of the wet polymeric membrane sample (cm^2^), and R is the measured membrane resistance obtained from the electrochemical impedance spectroscopy (EIS) data (S).

## 3. Results and Discussion

### 3.1. Laboratory Electrodialysis System Efficiency

The laboratory electrodialysis system was tested preliminary at different conditions (applied voltage at electrodes (3 V and 20 V)) and different concentration of wastewater containing zinc ions (between 0.5 g L^−1^ and 2.5 g L^−1^). The results showed that a lower applied voltage (3 V) and a low concentration of zinc ions (0.5 g L^−1^) leads to a low percentage removal of zinc ions. Moreover, a higher applied voltage (20 V) lead to increase of energy consumption. A concentrated solution (2.5 g L^−1^) leads to increased resistance of system, increased the energy consumption, destruction or damage to membranes, which go a long way to increasing the total costs. The electrodialysis system operated over 1 h, for treatment of wastewater containing zinc ions, led to a higher percentage removal of zinc ions, but the energy consumption was higher, and the prepared polymeric membranes showed cracks and finally decreases the membranes resistance. Due to these aspects mentioned above, which are not an advantage, in this study the laboratory electrodialysis system was chosen to be operated at values of applied voltages of 5 V, 10 V and 15 V, and a concentration of wastewater containing zinc ions of 2 g L^−1^. The calculated results of D_r_, P_r_, I_e_ and E_c_ for zinc ions, using Equations (1–4), after 1 h of treatment, for different values of applied voltage, are indicate in Table 2.

The data presented in Table 2 highlight that the values for D_r_ and P_r_ increase with increasing of the applied voltage at the lead electrodes. The higher values of D_r_ (73.93%) and of P_r_ (66.58%) were obtained for the polymeric membrane with SiO_2_ nanoparticles (C3), at 15 V, after 1 h of treatment. These higher values can be due to the enough current intensity that crossing the laboratory electrodialysis system that leads to a more intensive ionic migration. When the values of D_r_ and of P_r_ increase then the applied voltage increasing, and this fact can be due to migration of water that solvated the zinc ions. Also, this fact can be due to the increase of the differences on osmotic pressure between dilute (middle) and concentrate (anodic) compartments [14,35,36,37]. The increase of applied voltage led to a reduction in the dilute solution’s conductivity, due to the displacement of zinc ions through the polymeric membrane from the anodic compartment (concentrate solution) to the middle compartment (dilute solution) [35,36,37]. At applied voltage of 5 V, the values of P_r_ were lower (for sample F1 was 23.68% and for sample C1 was 39.64%) because the current that crossing the laboratory electrodialysis system (current between anode and cathode electrodes) was insufficient to facilitate the transport of ions through prepared polymeric membranes. Also, this can due to the concentration polarization in the polymeric membrane boundary layer as well as depletion of electron carriers in the middle compartment (dilute solution). The values of separation efficiency found in this work were in a good agreement with data provided in the literature [35,36,37]. Dalla Costa et al. [35] treated a metal finishing wastewater containing different metallic ions by electrodialysis and using Nafion and Selemion membranes (Nafion is a registered trademark of E. I. du Pont de Nemours and Co. and Selemion is a registered trademark of Asahi Glass Co.) [35]. They reported that the highest percent extraction of the zinc ions after 6 h of treatment under potentiostatic control was 20.3%. Babilas and Dydo [36] examined the possibility of selective zinc recovery from simulated electroplating industry wastes by electrodialysis enhanced with complex formation. The results indicated that the zinc recovery exceeded 86.6% with current efficiency equal to 84.95%, for heterogeneous ion-exchange membranes type CM(H)-AM(H) Ralex, at a constant voltage of 6 V, after 60 min of treatment.

The current efficiency was determined based on the values of the monitored current intensity using a power supply (AX-3005D, 0 ÷ 30 VDC, 0 ÷ 5 A, AXIOMET, Transfer Multisort Elektronik, Lodz, Poland). At the beginner of the electrodialysis process, the current intensity initial increase with time possible due to the heating of the solutions, and after 10 min decrease due to the resistance increase. The values of current efficiency decrease with the increasing of the applied voltage. The variation of current efficiency can be attributed to the decrease in the electrical resistance of the system. The lower value of I_e_ (2.65%) was obtained for polymeric membrane with SiO_2_ nanoparticles (sample C3). The increase in applied voltage decreases the current intensity of the system. This led to an increase of energy consumption. The higher value of E_c_ (147.33 kWh m^−3^) was obtained for polymeric membrane without SiO_2_ nanoparticles (sample F3) and the lower value of E_c_ (92.67 kWh m^−3^) was obtained for polymeric membrane with SiO_2_ nanoparticles (sample C3). This value was in a good agreement with data reported by other authors [35,37]. When high voltage (>15 V) is applied to electrodialysis system, containing high concentration of zinc solutions (>20 g L^−1^), the effect of concentration polarization becomes more prominent and current efficiency of electrodialysis significantly decreases [35].

At the end of each experiment, when the laboratory electrodialysis system was disassembled, the formation of wet metallic zinc layer deposited on the cathode electrode surface was observed and viewed using a camera (Panasonic LUMIX DMC-LS80, Panasonic AVC Networks Xiamen Co. Ltd., Xiamen, China) (Figure 4).

The metallic zinc deposits, after drying at room temperature (24 ± 1 °C), can be further used in various industry (e.g., galvanization, automobile, construction, light industry, machinery, chemical, pharmaceutical, batteries, pigments, textiles, paint) with different applications (e.g., covering material for roofs, zinc spelter, zinc wire, printing zinc plates, zinc-manganese batteries and zinc air batteries, rubber, it is suitable for making components and covers of instruments and meters). The dilute solution, obtained after the treatment using the laboratory electrodialysis system, can be released into the environment if has a concentration according to the quality standards and regulations set by the Environmental Protection Agency.

### 3.2. FTIR-ATR Spectroscopy

FTIR-ATR analysis was conducted to elucidate the nature and correlation between the polymer matrix chains and the incorporation of SiO_2_ nanoparticles (Figure 5).

The band observed between 3300–3500 cm^−1^ was attributed to the stretching O–H group, resulting from the intramolecular and intermolecular hydrogen bonds [39,40]. Bands in the range of 3020–2968 cm^−1^ correspond to the C–H stretching of –CH_2_ (samples F1–F3). The peak at 2944 cm^−1^ showed C–H stretching vibrations of PVA (samples F1–F3), while for samples C1–C3, C–H stretching vibration bands were observed at around 2935 cm^−^^1^. The peak has shifted towards lower wavenumber, after electrodialysis process, possible due to the carbonyl group interaction with hydrogen bonding. Also, it is possible that the free carbonyl group increased with the addition of silica particles [40]. The peaks that appear at 2244 cm^−1^ and ~1449 cm^−1^ can be assigned to C≡N stretching vibrations and –CH_2_ bending vibration, respectively [36,37,40]. The intensity of these peaks for membrane with SiO_2_ nanoparticles (C1–C3) increase after electrodialysis process. These results indicated the strong interactions of the incorporation of SiO_2_ nanoparticles in polymer matrix with zinc ions, confirmed also by the percentage removal of Zn^2+^. The peaks at 1370 cm^−1^ and around 1230 cm^−^^1^ can be due to symmetric vibrations of –CH_3_ and of C–O single bond stretching modes [37,39,40].

In all spectra of samples C0-C3, the peak that appear at ~1640 cm^−1^ can be assigned to C=C and corresponded to the bonded carbonyl groups, and the peak at 1737 cm^−1^ related to the free carbonyl groups [40,41,42]. The modification of carbonyl band, after electrodialysis process (1737 cm^−1^), confirmed the presence of hydrogen bonding between carbonyl and hydroxyl groups. The main characteristic absorption peaks of SiO_2_ were observed and were found at around 1078 cm^−1^ ascribed to Si–O–Si (asymmetric stretching vibration) and around 942 cm^−1^ ascribed to silanol (Si–OH) groups (bending vibration). The adsorption peaks at around 808 cm^−1^ and 457 cm^−1^ were ascribed to Si–O (symmetric stretching vibration) [41,42,43].

### 3.3. SEM Analysis

In order to study the effect of the incorporation of SiO_2_ nanoparticles in the structure of the prepared polymeric membranes, SEM micrographs at 2000× magnification of the top surface and cross-section polymeric membranes without and with SiO_2_ nanoparticles (samples F0 and C0), before used in the laboratory electrodialysis system, were examined (Figure 6a–d).

Figure 6a–d showed the top-surface and cross-section morphologies of polymeric membranes without (F0) and with (C0) SiO_2_ nanoparticles. The top-surface view of the polymeric membrane without SiO_2_ nanoparticles (sample F0) (Figure 6a) showed the formation of spongy like holes, more and large pores (between 25.26 µm and 60.67 µm), nodule and macro void like structure [42,43,44]. The top-surface morphology of the polymeric membrane (sample C0) (Figure 6c) was changed significantly with incorporation of SiO_2_ nanoparticles in the polymer matrix. The SEM images, for sample C0, showed the uniform distribution and dispersion of the SiO_2_ nanoparticles in the active layer of the prepared polymeric membrane. It can be observed that the SiO_2_ nanoparticles were regular on surface of polymeric membrane. This indicated that the SiO_2_ nanoparticles were fixed on the polymer main chain. Also, SEM images indicate that the nanoparticles are compatible with polymer matrix [37,41,42,43]. Moreover, the SiO_2_ nanoparticles incorporated in the prepared polymeric membrane, observed as white circular and non-circular shape, led to the formation of packed chain structure, reduce the pores size (between 1.045 µm and 288.5 nm) and reduce the macro void. The cross-section view of the polymeric membrane without SiO_2_ nanoparticles (Figure 6b) shows a fused nodule structure, whereas, the polymeric membrane with SiO_2_ nanoparticles had apparently loose nodule (Figure 6d). This can indicate the existence of interfacial stresses between SiO_2_ nanoparticles and polymer matrix. Figure 6d shows that the SiO_2_ nanoparticles are interconnected and densely incorporated and dispersed in the polymer matrix, which can be beneficial to improving the resistance and proton conductivity of the polymeric membrane [41,42]. The interaction between polymer matrix and SiO_2_ nanoparticles can prevent the silica nanoparticles agglomeration. The obtained results are in good agreement with other data reported in literature [42,43]. Petcu et al. [37] showed that the introduction of fumed silica powder into the polymer matrix structure of polymeric membrane reduces the pore size. Jia et al. [41] studied the polyvinyl alcohol/silica nanocomposites derived from copolymerization of vinyl silica nanoparticles and vinyl acetate. They showed that the silica nanoparticles and PVA matrix were well compatible. Also, the size of silica nanoparticles dispersed in PVA matrix remained between 30 and 40 nm and almost equal. This suggested that the copolymerization had taken place.

### 3.4. EIS Measurements

The electrochemical impedance spectroscopy (EIS) is a versatile electrochemical technique used to evaluate the electrical response of the polymeric membranes (e.g., resistance of electron charge transfer, double layer capacitance, electrical resistance (impedance), conductivity) [14,38].

The electrochemical impedance measurements have been carried out at room temperature (24 ± 1 °C). The Bode diagrams for all polymeric membranes (without (F0–F3) and with (C0–C3) SiO_2_ nanoparticles), before and after electrodialysis tests) (Figure 7 and Figure 8), were realized to indicate the relation between the impedance (Z), the frequency and the phase angle.

Figure 7 and Figure 8 show that the impedance decrease with increasing of frequencies possible due to a capacitive resistance change transfer at the electrode–wet membrane interface that is in series with the impedance of the cell elements. It can be observed that the system impedance (at the limit of zero frequency) decrease as applied voltage increased. This can be due to the decreases of interfacial ionic charge transfer from the solution phase through the electric double layer to the membrane and metallic ions that migrated through the interfacial double layers [14,37,38]. Due to the introduction of SiO_2_ nanoparticles in the polymer matrix, the strength of the static electric interaction between the Zn^2+^ and the negatively charged fixed groups (e.g., –SO_3_H, –CH_3_ or –CH_2_–) can increased and can lead in an increase in ion transfer resistance [44,45].

Proton conductivity (σ) of the polymeric membranes without and with SiO_2_ nanoparticles was calculated from the bulk resistance that was determined from the high frequency intercept of the imaginary component of the impedance with the real axis. The calculated proton conductivities are presented in Figure 9.

It can be observed from Figure 9 that the proton conductivity of the polymeric membranes increases with the increasing of applied voltage. The proton conductivities of all polymeric membrane samples were higher than 0.3 × 10^−3^ S cm^−1^. For the initial polymeric membrane with SiO_2_ nanoparticles (sample C0), the value of proton conductivity was 0.642 × 10^−3^ S cm^−1^ and for the polymeric membrane without SiO_2_ nanoparticles (sample F0), the value of proton conductivity was 0.366 × 10^−3^ S cm^−1^. This shows that the presence of SiO_2_ nanoparticles in the polymeric membranes influences the electrical conductivity that have an important role in the metallic ion transport through the polymeric membranes. The increase of values of the proton conductivity of polymeric membranes after electrodialysis process, from 0.381 × 10^−3^ S cm^−1^ to 0.471 × 10^−3^ S cm^−1^ (sample F1–F3) and from 0.678 × 10^−3^ S cm^−1^ to 0.840 × 10^−3^ S cm^−1^ (sample C1–C3), can be due to the increases of the mobility of zinc ions and polymer chains. An increase in the value of the proton conductivity can be attributed to the increases of the mobility of free ions as applied voltage was increasing. The higher value of proton conductivity was obtained for the polymeric membrane with SiO_2_ nanoparticles (sample C3) of 0.840 × 10^−3^ S cm^−1^. This can be attributed to the more zinc ions transport through the polymeric membrane and due to the incorporation of SiO_2_ nanoparticles in polymer matrix. The obtained results are in concordance with the values obtained for percentage removal of zinc ions. The final values of proton conductivity of the prepared polymeric membrane in present work is acceptable at the current stage and are in good agreement with other data reported in literature [45,46,47]. Ahmadian-Alam et al. [46] prepared the nanocomposite polymer electrolyte membrane contained sulfonic acid functionalized silica and imidazole-encapsulated NH_2_-MIL-53(Al) (MOF) nanoparticles. They reported that the proton transport of the prepared nanocomposite membrane increases to 0.017 × 10^−^^3^ S cm^−1^ by adding only 5% of sulfonic acid functionalized silica and imidazole-encapsulated MOF. Zuo et al. [47] developed a poly(vinylidene fluoride)/ethyl cellulose and amino-functionalized nano-SiO_2_ (PVDF-EC-(A-SiO_2_)) composite coated on polyethylene. They related that the higher proton conductivity of the composite membrane was of 0.79×10^−^^3^ S cm^−1^. The data reported in the literature reveals that the higher proton conductivity could be obtained by introducing much more functional groups (e.g., –SO_3_H, R-N(CH_3_)_3_OH, R-SO_3_^−^) into the polymer chains [44,47].

## 4. Conclusions

In this paper, the polymeric membranes without and with SiO_2_ nanoparticles were successfully prepared and applied for the removal of zinc ions (Zn^2+^) from a simulated wastewater using a new versatile laboratory electrodialysis system. The experimental results from the electrodialysis process indicated that the values of demineralization rate and of percentage removal of zinc ions increase with the increasing of applied voltage. This can be due to migration of water that solvated the zinc ions. The results revealed that the higher value of percentage removal of zinc ions (66.58%) was obtained for the polymeric membrane with SiO_2_ nanoparticles, at higher applied voltage (15 V), after 1 h of treatment. FTIR-ATR spectra showed the strong interactions of SiO_2_ nanoparticles incorporated in polymer matrix with zinc ions. SEM micrographs of the top-surface and cross-section of polymeric membrane with SiO_2_ nanoparticles indicated that the SiO_2_ nanoparticles were uniform distributed and dispersed in the polymer matrix of the prepared polymeric membrane. The values of proton conductivity for all polymeric membranes increase with the increasing of applied voltage. The impedance measurements for proton conductivity of polymeric membranes with SiO_2_ nanoparticles suggested a considerable change.

The results demonstrated that the synthesized polymeric membrane is excellent for efficient removal of metallic ions from wastewater and can be used for future applications (e.g., fuel cells, nanofiltration, electrodialysis reverse, ultrafiltration, reverse osmosis, microfiltration). Also, the new versatile laboratory electrodialysis system can future applied for water desalination, treatment of industrial effluents and removal/recover of other metallic ions (e.g., Ni^2+^, Cu^2+^, Fe^2+^, Cd^2+^, Pb^2+^) from different wastewaters.

## Figures and Tables

**Figure 1 polymers-13-01875-f001:**
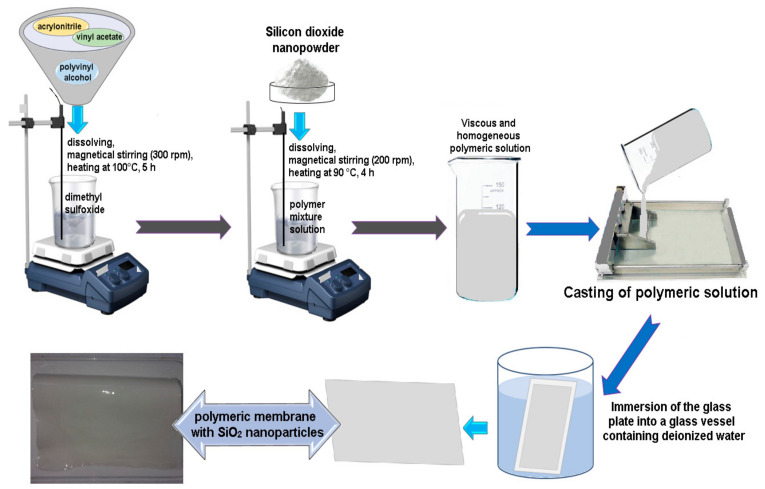
Schematic representation of the preparation polymeric membrane with SiO_2_ nanoparticles.

**Figure 2 polymers-13-01875-f002:**
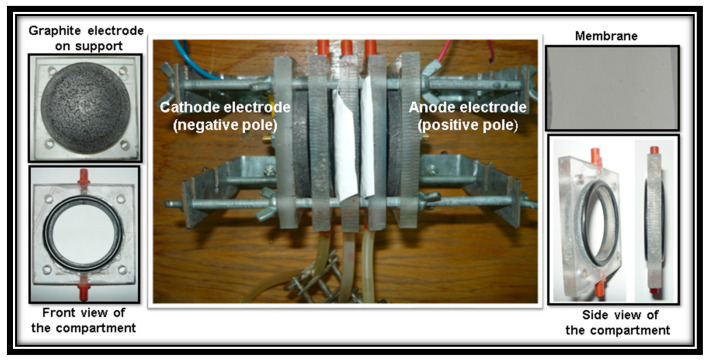
Laboratory electrodialysis system with polymeric membranes.

**Figure 3 polymers-13-01875-f003:**
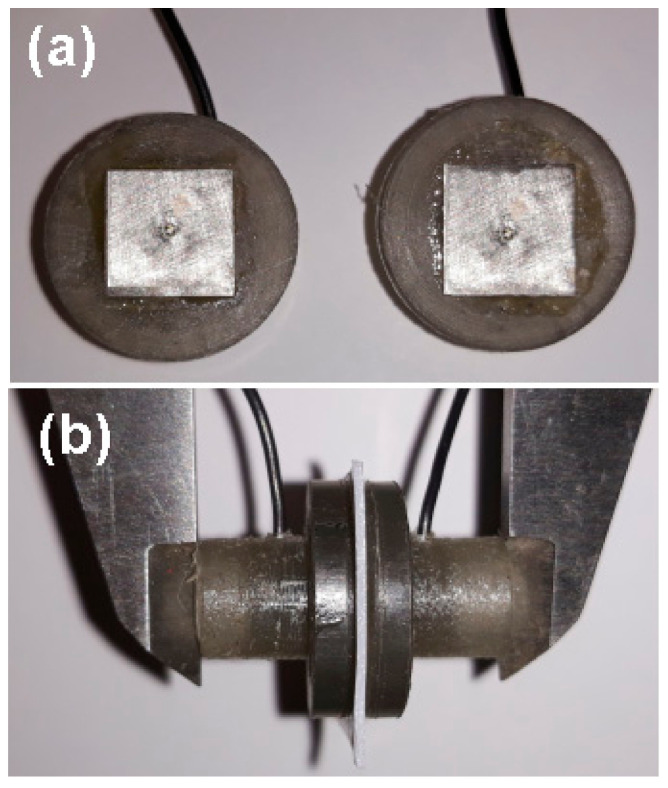
Electrochemical cell for EIS measurements: (**a**) two symmetrical platinum electrodes and (**b**) small sample of wet prepared polymeric membrane placed between platinum electrodes.

**Figure 4 polymers-13-01875-f004:**
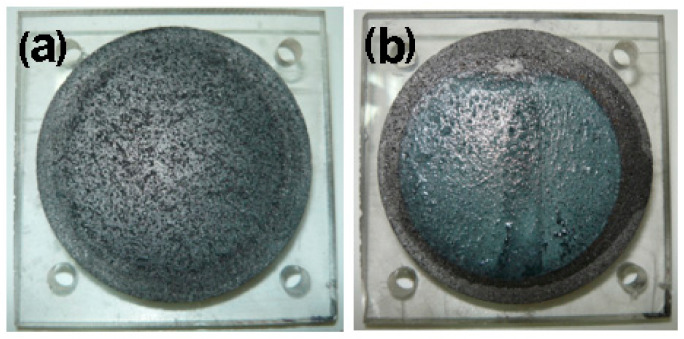
Image of cathode electrode surface before (**a**) and after (**b**) metallic zinc layer deposited.

**Figure 5 polymers-13-01875-f005:**
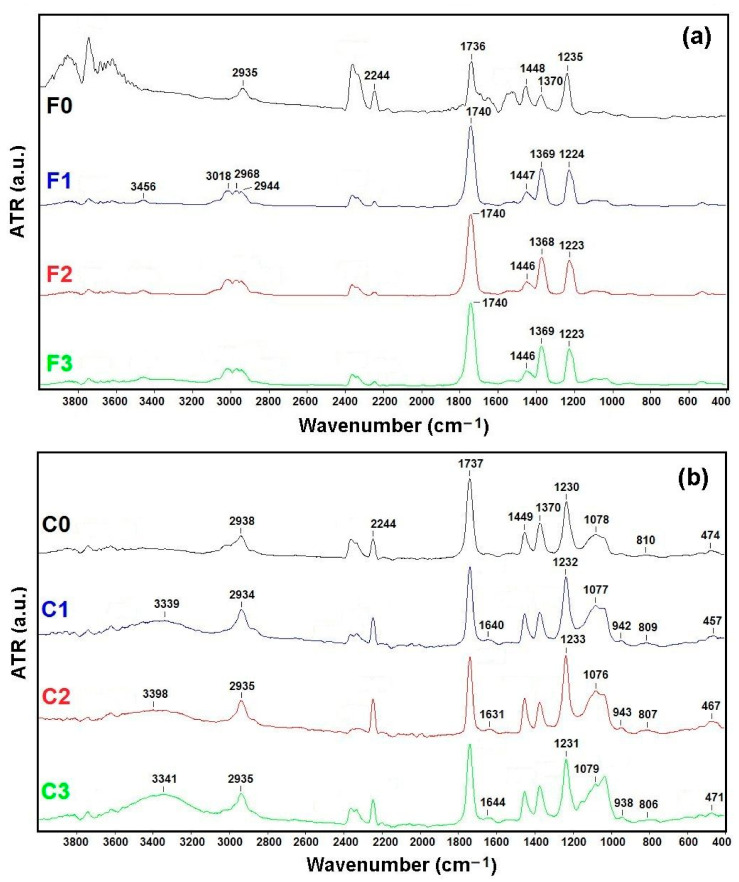
FTIR-ATR spectra for polymeric membranes without (**a**) and with (**b**) SiO_2_ nanoparticles, before (F0 and C0) and after (F1–F3 and C1–C3) 1 h of treatment.

**Figure 6 polymers-13-01875-f006:**
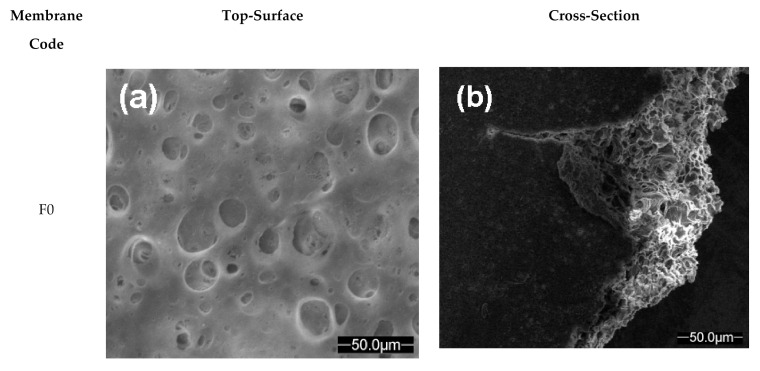
SEM images of polymeric membranes (without and with SiO_2_ nanoparticles): F0, top-surface (**a**) and cross-section (**b**); C0, top-surface (**c**) and cross-section (**d**).

**Figure 7 polymers-13-01875-f007:**
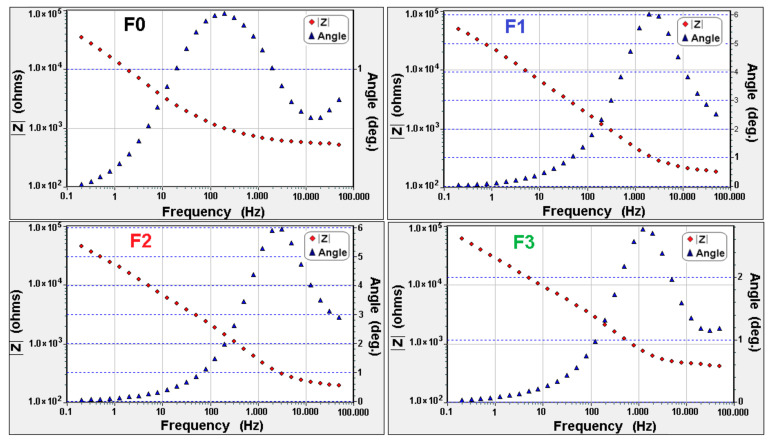
Bode diagrams for polymeric membranes without SiO_2_ nanoparticles, before (F0) and after (F1–F3) 1 h of treatment.

**Figure 8 polymers-13-01875-f008:**
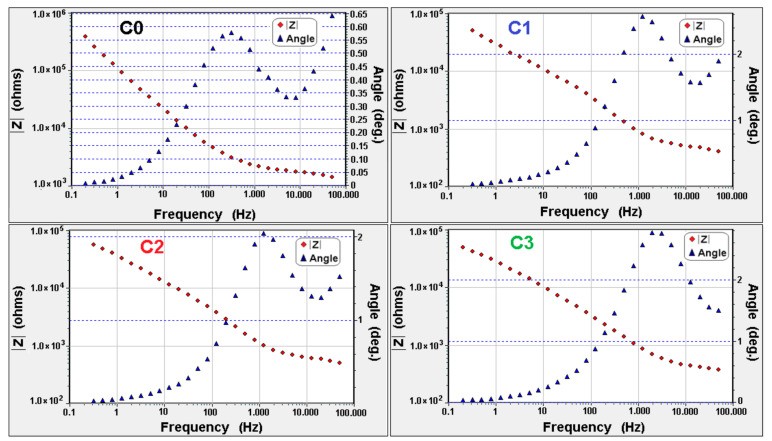
Bode diagrams for polymeric membranes with SiO_2_ nanoparticles, before (C0) and after (C1–C3) 1 h of treatment.

**Figure 9 polymers-13-01875-f009:**
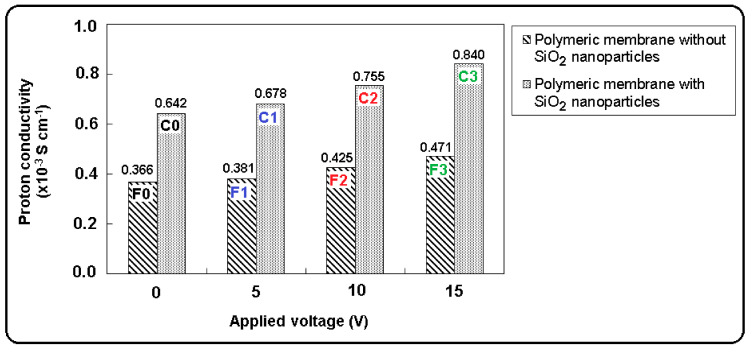
Proton conductivity values of polymeric membranes without and with SiO_2_ nanoparticles, before and after used in the electrodialysis system.

**Table 1 polymers-13-01875-t001:** Codes of polymeric membranes without and with SiO_2_ nanoparticles, before (applied voltage: 0 V) and after electrodialysis process (applied voltage: 5, 10 and 15 V).

Code	Membrane Type	Applied Voltage, V
F0	Polymeric membrane without SiO_2_ nanoparticles	0
F1		5
F2		10
F3		15
C0	Polymeric membrane with SiO_2_ nanoparticles	0
C1		5
C2		10
C3		15

**Table 2 polymers-13-01875-t002:** D_r_, P_r_, I_e_ and E_c_ values obtained after 1 h of treatment.

Membrane Code	D_r_, %	P_r_, %	I_e_, %	E_c_, kWh m^−3^
F1	27.93	23.68	23.22	5.55
F2	40.41	44.63	11.01	26.83
F3	66.28	58.63	4.73	147.33
C1	36.48	39.64	10.70	4.17
C2	49.31	53.61	6.62	23.93
C3	73.93	66.58	2.65	92.67

## Data Availability

Not applicable.

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
