# Peer review of "Study of Polyvinyl Alcohol-SiO2 Nanoparticles Polymeric Membrane in Wastewater Treatment Containing Zinc Ions"

_polymers, 2021, doi:10.3390/polym13111875_

Round 1

Reviewer 1 Report

This manuscript reported a very high performance PVA-SiO2 membrane compared to the studies (Table 5) without providing in-depth characterization and discussion. Looking at the Ref [60] in Table 5, the work also used PVA-SiO2 membrane but reported 10 times lower proton conductivity compared to the findings reported by authors. It is completely unknown on how both similar membranes can show such large difference in findings.

In addition, the authors claimed the membrane is “novel” is completely misleading. There are many studies also utilized PVA-SiO2 in making membranes!!

The organization of the entire is bad. Besides many grammatical mistakes and bad constructed sentences, the authors included many unnecessary photographs (Figure 1, 2, 3 and 4) which are nothing related to the findings in this manuscript.

In addition, the inclusion of a summary table which covered many different membrane processes are NOT necessary. This is a technical paper and NOT a review article.

Also, the arrangement of equations (Equation 1-4) in the Results and Discussion section is completely wrong. Such equations should appear in Methodology section.

The research gap on why this research needed to carry out is not clear and in fact, I have no ideal why it is so important and needs to be carried out in the first place.

Figure 5 and 6 – There are too many peaks indicated by the authors. Authors should emphasize the peaks that can differentiate among the samples. Besides, both figures should be arranged in one figure.

Figure 7 – Only selective images (two membranes) are provided in this manuscript. This is not sufficient to draw a conclusion. Besides, the cross-sectional morphology of membrane should be presented and correlate it with transport resistance.

Figure 8’s caption is about C0-C3 membranes, but the results shown are F0-F3 membrane. I strongly advise the authors to arrange all the data within 1 image so as we can clearly see the differences! Same comments go to Figure 9.

Lastly, it is lack of in-depth characterization to justify the results. XPS and leaching test on the SiO2 should be performed! Both are compulsory!

Author Response

Response to Reviewer 1 Comments

Point 1. This manuscript reported a very high performance PVA-SiO2 membrane compared to the studies (Table 5) without providing in-depth characterization and discussion. Looking at the Ref [60] in Table 5, the work also used PVA-SiO2 membrane but reported 10 times lower proton conductivity compared to the findings reported by authors. It is completely unknown on how both similar membranes can show such large difference in findings.

Response 1: Thank you for your remark.

Ref [60] is in the corrected manuscript Ref [59].

In the Ref [59] the EIS analysis was performed on a VoltaLab 40 Dynamic Electrochemical System used in conjunction with associated data processing software, VoltaMaster 4. The impedance measurements were carried out using an electrochemical cell consisting of two plan-parallel stainless steel disk electrodes containing in-between vices the membranes subjected to the evaluation. In order to prepare the polymer membranes, was used 10% wt. fumed silica powder.  

In present paper, all electrochemical impedance experiments have been carried using a potentiostat/galvanostat (Parstat CS350 Electrochemical System, Wuhan Corrtest Instruments Corp. Ltd., China), associated with an integrated software (ZSimpWin version 3.5). The EIS measurements were performed using an electrochemical cell composed by two symmetrical platinum electrodes. In order to prepare the polymeric membranes was used 15% wt. SiO2 nanoparticles (silicon dioxide nanopowder (SiO2, 10÷20 nm size)).

The electrochemical impedance spectroscopy (EIS) method is sensitive to fouling at multiple interfaces of the membrane (outer surface, active and support layers).

The EIS data are influenced by the device/equipment and the electrodes used to determine the proton conductivity of the membranes. For this reason, the results obtained in this study are different from those in the literature.

Point 2. In addition, the authors claimed the membrane is “novel” is completely misleading. There are many studies also utilized PVA-SiO2 in making membranes!!

Response 2: Thank you for your remark.

The word “novel” was changed with “new”, for more clarity.

Point 3. The organization of the entire is bad. Besides many grammatical mistakes and bad constructed sentences, the authors included many unnecessary photographs (Figure 1, 2, 3 and 4) which are nothing related to the findings in this manuscript.

Response 3: Thank you for your remark and suggestions.

The manuscript was reorganized, for a better clarity and understanding of the experiments performed.

English has been improved and grammatical and expression errors have been corrected.

Certain figures have been corrected and highlighted in the text of the manuscript. These cannot be removed from the manuscript because we want to highlight what was used in this study, not just words. Moreover, clarity is also desired for the readers of the manuscript.

Point 4. In addition, the inclusion of a summary table which covered many different membrane processes are NOT necessary. This is a technical paper and NOT a review article.

Response 4: Thank you for your remark.

Table 1 and Table 5 have been removed and replaced with explanations.

In order not to create confusion between a manuscript and a review, the most relevant and important information was included.

Electrodialysis has been successfully applied at the laboratory scale and at the industrial level for treated waters and wastewaters containing contaminants and products such as: heavy metal ions [20,25-27], acids [28], dyes [26,29], organic matter [28,30]. The wastewaters may come from different industries such as: chemistry [25,27,30], (bio)chemistry [31], food processing [32], pharmaceutical [27,31], mining and refining [21,25,26, 32,33]. Electrodialysis was successfully applied for the removal of contaminants from different waters and wastewaters due to their advantages such as: high separation selectivity and efficiency, does not require specialized equipment, operates without noise, low maintenance, low capital cost, does not require specialized equipment, operates without noise, does not require the application of pressure, low space and material requirements [20,29-31,35-37].

The final values of proton conductivity of the prepared polymeric membrane in present work is acceptable at the current stage and are in good agreement with other data reported in literature [71-73]. Ahmadian-Alam et al. [71] prepared the nanocomposite polymer electrolyte membrane contained sulfonic acid functionalized silica and imidazole-encapsulated NH2-MIL-53(Al) (MOF) nanoparticles. They reported that the proton transport of the prepared nanocomposite membrane increases to 0.017×103 S cm-1 by adding only 5% of sulfonic acid functionalized silica and imidazole-encapsulated MOF. Zuo et al. [72] developed a poly(vinylidene fluoride)/ethyl cellulose and amino-functionalized nano-SiO2 (PVDF-EC-(A-SiO2)) composite coated on polyethylene. They related that the higher proton conductivity of the composite membrane was of 0.79×103 S cm-1.

Point 5. Also, the arrangement of equations (Equation 1-4) in the Results and Discussion section is completely wrong. Such equations should appear in Methodology section.

Response 5: Thank you for your suggestion.

The equations 1-4 were introduced in the Methodology section.

Point 6. The research gap on why this research needed to carry out is not clear and in fact, I have no ideal why it is so important and needs to be carried out in the first place.

Response 6: Thank you for your information.

New information was introduced throughout the manuscript to highlight the importance of the study conducted both for researchers and for students, masters, doctoral students, who conduct research about this manuscript.

Point 7. Figure 5 and 6 – There are too many peaks indicated by the authors. Authors should emphasize the peaks that can differentiate among the samples. Besides, both figures should be arranged in one figure.

Response 7: Thank you for your suggestion.

Figure 5 and Figure 6 were changed and arranged in one figure. Moreover, only the relevant peaks were highlighted and indicated, which highlight the differences between the samples.

Point 8. Figure 7 – Only selective images (two membranes) are provided in this manuscript. This is not sufficient to draw a conclusion. Besides, the cross-sectional morphology of membrane should be presented and correlate it with transport resistance.

Response 8: Thank you for your suggestion.

Figure 7 was changed.

The cross-sectional morphology of membrane is presented and correlate it with transport resistance. Also, more information was included.

Figure 6a-d showed the top-surface and cross-section morphologies of polymeric membranes without (F0) and with (C0) SiO2 nanoparticles.

The cross-section view of the polymeric membrane without SiO2 nanoparticles (Figure 6b) shows a fused nodule structure, whereas, the polymeric membrane without SiO2 nanoparticles had apparently loose nodule (Figure 6d). This can indicate the existence of interfacial stresses between SiO2 nanoparticles and polymer matrix. Figure 6d shows that the SiO2 nanoparticles are interconnected and densely incorporated and dispersed in the polymer matrix, which can be beneficial to improving the resistance and proton conductivity of the polymeric membrane

Point 9. Figure 8’s caption is about C0-C3 membranes, but the results shown are F0-F3 membrane. I strongly advise the authors to arrange all the data within 1 image so as we can clearly see the differences! Same comments go to Figure 9.

Response 9: Thank you for your remark.

The title of the Figure was corrected.

Because Figure 5 and Figure 6 were arranged in one figure, the number of Figure 8 and Figure 9 were changed. In the manuscript these figures were not changed, because the program used, which is licensed, does not allow the overlapping of evidence. Moreover, if this were done, as there is a lot of data, you may not notice very clearly the differences between the samples. You can import data and make graphs through a program without a license, or in excel or origin. It was wanted that the manuscript be presented as clearly as possible, that the figures to be very clear and visible. Overlapping samples in a single graph leads to blurred and very crowded graphs curves.

Point 10. Lastly, it is lack of in-depth characterization to justify the results. XPS and leaching test on the SiO2 should be performed! Both are compulsory!

Response 10: Thank you for your suggestion.

The aim of this study was to obtain polymeric membranes containing polyvinyl alcohol-SiO2 nanoparticles and to observe their effectiveness in the process of removing zinc ions through the electrodialysis process.

Also, the XPS and leaching test on the SiO2 cannot be performed and included in the present manuscript, because is the subject of another manuscript that will be published in another Journal.

The English was improved and corrected.

Reviewer 2 Report

The paper discusses: ‘’Study of Polyvinyl Alcohol-SiO2 Nanoparticles Polymeric Membrane in Wastewater Treatment Containing Zinc Ions’’, is well written with the idea being clearly explained by the authors. I would also like to comment that the title be modified a little. Below are few suggestions to the authors:

  1. Line 47, sentence needs a construction, please rewrite the sentence,
  2. Line 57, please rewrite it, the sentence needs construction, the same applies for line 59
  3.  I would prefer to use people/person(s) in line 62
  4. Line 70 to 75 can be reconstructed better, instead of brackets , words like “such as’’ and “including” may be utilized
  5. Please provide clarity in the last paragraph in Table 1 under limitations, is it decrease or de-crease
  6. Line 89, sentence construction, please rewrite the sentence
  7. Line 90-94, please rewrite the paragraph
  8. Line 95-100, I don’t understand what is the purpose of such a paragraph, but surely it cannot be part of the introduction, it looks like someone was commenting on the document or something
  9. Check line 134, sentence construction
  10. Line 157, please rewrite it again
  11. Line 159, please rewrite it again, sentence construction
  12. I think polyvinyl alcohol preparation can be a section rather than such a preparation being incorporated in the material section.
  13. Line 162, sentence construction
  14. Line 170, sentence construction
  15. Magnetically could be simplified to magnetic stirring
  16. What could be a possible reason for low removal of Zn ions at low applied voltage an Zn ions concentration
  17. Line 295 needs a sentence construction
  18. Line 299, could be put as: The increase in applied voltage…..
  19. Line 460, sentence construction is needed, same for line 465
  20. In the conclusion section, are there any future recommendations for this membrane system, based on the results

Author Response

Response to Reviewer 2 Comments

The paper discusses: ‘’Study of Polyvinyl Alcohol-SiO2 Nanoparticles Polymeric Membrane in Wastewater Treatment Containing Zinc Ions’’, is well written with the idea being clearly explained by the authors. I would also like to comment that the title be modified a little. Below are few suggestions to the authors:

Point 1. Line 47, sentence needs a construction, please rewrite the sentence,

Response 1: Thank you for your remark.

The sentence has been rewritten.

The European Environment Protection Agency and the World Health Organization established lists with the dangerous substances and inorganic pollutants where zinc has been included [2,4,5]. Based on the lists, the World Health Organization, Environment Protection Agency, Ministry of Environment and Forest, established limits of discharge of zinc from industrial activities into the environment as follows: for estuaries and marine waters of 40 µg L-1, for freshwater at 45÷500 µg L-1 and for wastewater of 5 mg L-1 [3–5].

Point 2. Line 57, please rewrite it, the sentence needs construction, the same applies for line 59

Response 2: Thank you for your remark.

The sentence has been rewritten.

The World Health Organization and The Institute of Medicine recommended as a tolerable level of zinc from foods of 15 mg day-1 for adults [9,10]. The tolerable level of zinc from supplements was established for an adult of 40 mg day-1 [10] and for a children of 2 mg day-1.

Point 3. I would prefer to use people/person(s) in line 62

Response 3: Thank you for your suggestion.

The word “humans” was changed with word “persons”.

Point 4. Line 70 to 75 can be reconstructed better, instead of brackets , words like “such as’’ and “including” may be utilized

Response 4: Thank you for your suggestion and recommendation.

Lines were reconstructed better.

Because these methods have limitations, such as: large amount of sludge, extra operational cost for sludge disposal [15,18], large amount of chemicals for removal of metals from waters and wastewaters [15,16], addition of reagents and coagulants [16,17], involves large activation energy [18], low selectivity [19], membrane technologies, such as: electrodialysis [20,21], ultrafiltration [22,23], reverse osmosis [23], nanofiltration [24], has been successfully applied at the laboratory scale and at the industrial level.

Point 5. Please provide clarity in the last paragraph in Table 1 under limitations, is it decrease or de-crease

Response 5: Thank you for your suggestion and remark.

Table 1 was eliminated and was included only relevant and important information.

Because these methods have limitations, such as: large amount of sludge, extra operational cost for sludge disposal [15,18], large amount of chemicals for removal of metals from waters and wastewaters [15,16], addition of reagents and coagulants [16,17], involves large activation energy [18], low selectivity [19], membrane technologies, such as: electrodialysis [20,21], ultrafiltration [22,23], reverse osmosis [23], nanofiltration [24], has been successfully applied at the laboratory scale and at the industrial level. Each technique has its advantages and limitations. The most significant and relevant advantages of membrane technologies, regarding the treatment of polluted waters and wastewaters are: high efficiency, can function at low temperature, additional chemicals are only needed in very small amounts or are not needed for certain membrane processes (e.g., electrodialysis, nanofiltration), low environmental impact [20,22,24].

Point 6. Line 89, sentence construction, please rewrite the sentence

Response 6: Thank you for your remark.

The sentence has been rewritten.

At the final of the electrodialysis process with three compartments, as a result of the transport process, the solution from the central compartment will be diluted in pollutants, and the solution from the anodic compartment will be enriched/concentrated in pollutants.

Point 7. Line 90-94, please rewrite the paragraph

Response 7: Thank you for your remark.

The paragraph was rewrite.

Electrodialysis has been successfully applied at the laboratory scale and at the industrial level for treated waters and wastewaters containing contaminants and products such as: heavy metal ions [20,25-27], acids [28], dyes [26,29], organic matter [28,30]. The wastewaters may come from different industries such as: chemistry [25,27,30], (bio)chemistry [31], food processing [32], pharmaceutical [27,31], mining and refining [21,25,26, 32,33]. Electrodialysis was successfully applied for the removal of contaminants from different waters and wastewaters due to their advantages such as: high separation selectivity and efficiency, does not require specialized equipment, operates without noise, low maintenance, low capital cost, does not require specialized equipment, operates without noise, does not require the application of pressure, low space and material requirements [20,29-31,35-37].

Point 8. Line 95-100, I don’t understand what is the purpose of such a paragraph, but surely it cannot be part of the introduction, it looks like someone was commenting on the document or something

Response 8: Thank you for your remark.

The paragraphs are come from template of the manuscript and have not been inadvertently deleted.

The paragraphs were deleted: “The introduction should briefly place the study in a broad context and highlight why it is important. It should define the purpose of the work and its significance. The current state of the research field should be carefully reviewed and key publications cited. Please highlight controversial and diverging hypotheses when necessary. Finally, briefly mention the main aim of the work and highlight the principal conclusions. As far as possible, please keep the introduction comprehensible to scientists outside your particular field of research.”

Point 9. Check line 134, sentence construction

Response 9: Thank you for your suggestion.

The sentence has been rewritten.

The results showed that the maximum values of removal percentage were of 74.8% for Cd and of 64.5% for Sn.

Point 10. Line 157, please rewrite it again

Response 10: Thank you for your remark.

The paragraph was rewrite.

Polyvinyl alcohol solution, due to stabilization effect, was used for the preparation of polymeric membranes. This was prepared in the laboratory by following procedure: firstly, a solution of polyvinyl acetate was obtained by radical polymerization of vinyl acetate in methanol

Point 10. Line 159, please rewrite it again, sentence construction

Response 10: Thank you for your remark.

The paragraph was rewrite.

Polyvinyl alcohol solution, due to stabilization effect, was used for the preparation of polymeric membranes. This was prepared in the laboratory by following procedure: firstly, a solution of polyvinyl acetate was obtained by radical polymerization of vinyl acetate in methanol (80:20, wt.%), at 60 °C, under refluxing, using as initiator the 2,2-Azobis(2-methylproprionitrile) (98%, Sigma-Aldrich, Merck KGaA, Darmstadt, Germany). After that, to obtain the final solution of polyvinyl alcohol, a mixture of polyvinyl acetate and methanol (40%) was catalyzed in the presence of 2% of sodium hydroxide (NaOH) (Merck, Redox Lab Supplies Com SRL, Bucharest, Romania) (calculated to amount of polyvinyl acetate), at 50 °C. Vinyl acetate and methanol were supplied by Sigma-Aldrich (Merck, Redox Lab Supplies Com SRL, Bucharest, Romania).

Point 11. I think polyvinyl alcohol preparation can be a section rather than such a preparation being incorporated in the material section.

Response 11: Thank you for your suggestion.

The polyvinyl alcohol preparation was included in the membrane preparation section.

Point 12. Line 162, sentence construction

Response 12: Thank you for your recommendation.

The sentence has been rewritten.

After that, to obtain the final solution of polyvinyl alcohol, a mixture of polyvinyl acetate and methanol (40%) was catalyzed in the presence of 2% of sodium hydroxide (NaOH)

Point 13. Line 170, sentence construction

Response 13: Thank you for your recommendation.

The sentence has been rewritten.

The polyvinyl alcohol-SiO2 nanoparticles polymeric membrane was obtained by wet-phase inversion method at room temperature (24 ± 1 °C), by following procedure: in a beaker was added a mixture of 3.5 g of copolymers (acrylonitrile (C3H3N) (70%) and vinyl acetate (C4H6O2) (30%)), and polyvinyl alcohol solution (20%) was completely dissolved in 50 mL of dimethyl sulfoxide ((CH3)2SO)

Point 14. Magnetically could be simplified to magnetic stirring

Response 14: Thank you for your suggestion.

Magnetically was simplified to magnetic stirring.

Point 15. What could be a possible reason for low removal of Zn ions at low applied voltage an Zn ions concentration

Response 15: Thank you for your question.

More information and explanations were included in the manuscript.

The applied cell voltage is a critical operating condition in electrodialysis processes as the voltage determines the current in the cell and hence the electrodialysis efficiency as well as energy consumption.

At applied voltage of 5 V the values of Pr were lower (for sample F1 was 23.68% and for sample C1 was 39.64%) because the current that crossing the laboratory electrodialysis system (current between anode and cathode electrodes) was insufficient to facilitate the transport of ions through prepared polymeric membranes. Also, this can due to the concentration polarization in the polymeric membrane boundary layer as well as depletion of electron carriers in the middle compartment (dilute solution).

According to the classical theory of current polarization for ion-exchange membranes, the current should increase linearly at low voltage, then increase more slowly and finally reach a “plateau” characterizing the so-called limiting current. The flux of ions removed by this process is limited by the concentration polarization at the interfaces between membranes and solutions

Point 15. Line 295 needs a sentence construction

Response 15: Thank you for your suggestion.

These higher values can be due to the enough current intensity that crossing the laboratory electrodialysis system that lead to a more intensive ionic migration.

Point 16. Line 299, could be put as: The increase in applied voltage…..

Response 16: Thank you for your suggestion.

“The increase of applied voltage …” was chanced with “The increase in applied voltage…”

Point 17. Line 460, sentence construction is needed, same for line 465

Response 17: Thank you for your suggestion.

The sentence has been rewritten.

In this paper, a new polymeric membrane without and with SiO2 nanoparticles was successfully prepared and applied for the removal of zinc ions (Zn2+) from a simulated wastewater using a new versatile laboratory electrodialysis system. Electrodialysis experiments confirmed suitability and efficiency of the obtained polymeric membrane for removal of Zn2+ by determination of demineralization rate and percentage removal of zinc ions. Also, the performance of laboratory electrodialysis system was evaluated by calculating the current efficiency (Ie) and energy consumption (Ec). The experimental results from the electrodialysis process indicated that the values of demineralization rate and of percentage removal of zinc ions increase with the increase of applied voltage.

Point 18. In the conclusion section, are there any future recommendations for this membrane system, based on the results

Response 18: Thank you for your suggestion.

In the conclusion section was included future recommendations.

The results suggest that the synthesized new polymeric membrane is excellent for efficient removal of metallic ions from wastewater and show promise for future applications (e.g., fuel cells, nanofiltration, electrodialysis reverse, ultrafiltration, reverse osmosis, microfiltration). Also, the new versatile laboratory electrodialysis system can future applied for water desalination, treatment of industrial effluents and removal/recover of other metallic ions, from different wastewaters, such as: Ni2+, Cu2+, Fe2+, Cd2+, Pb2+.

Reviewer 3 Report

The manuscript is interesting. In particular, the methodology for obtaining PVA membranes is novel. However, many points must be corrected.

1) The conclusions should be rewritten.

2) In the introduction several works are cited on membranes for pollutant removal that are not related to work. In particular, ceramic membranes. Instead, publications in which PVA membranes are used to remove arsenic or other contaminants are omitted. See for example:

Reversible swelling as a strategy in the development of smart membranes from electrospun polyvinyl alcohol nanofiber mats

Journal of Polymer Science  58(5), pp. 737-746 (2020)

Enhancing arsenic adsorption via excellent dispersion of iron oxide nanoparticles inside poly(vinyl alcohol) nanofibers

Journal of Environmental Chemical Engineering, Volume 9, Issue 1,February 2021, 104664

Biohybrid membranes for effective bacterial vehiculation and simultaneous removal of hexavalent chromium (CrVI) and phenol

Applied Microbiology and Biotechnology volume 105, pages827–838 (2021)

These works should be cited.

3) The authors say: “the surface morphologies of the polymeric membrane (sample C0) were changed significantly with addition of SiO2 nanoparticles. However, is this due to the addition of SiO2, or is it due to the modifications produced by the process carried out to incorporate the nanoparticles? . Please explain.

4) The morphology of the composite material (C0) indicates that the material is anisotropic. What is the reason for the orientation that was generated, taking into account that the starting material is isotropic?

5) What is the adhesion of the SI02 nanoparticles to the matrix? Adhesion tests should be carried out.

6) The authors show that indeed the membrane has the capacity to ad / absorb Zn2 +. What is the adsorption rate? How much is the adsorption capacity per gram of membrane?

Author Response

Response to Reviewer 3 Comments

The manuscript is interesting. In particular, the methodology for obtaining PVA membranes is novel. However, many points must be corrected.

Point 1: The conclusions should be rewritten.

Response 2: Thank you for your suggestion.

Conclusion was improved and rewritten.

In this paper, a new polymeric membrane without and with SiO2 nanoparticles was successfully prepared and applied for the removal of zinc ions (Zn2+) from a simulated wastewater using a new versatile laboratory electrodialysis system. Electrodialysis experiments confirmed suitability and efficiency of the obtained polymeric membrane for removal of Zn2+ by determination of demineralization rate and percentage removal of zinc ions. Also, the performance of laboratory electrodialysis system was evaluated by calculating the current efficiency (Ie) and energy consumption (Ec). The experimental results from the electrodialysis process indicated that the values of demineralization rate and of percentage removal of zinc ions increase with the increase of applied voltage. This can be due to migration of water that solvated the zinc ions. The results revealed that the higher value of percentage removal of zinc ions (66.58%) was obtained for the polymeric membrane with SiO2 nanoparticles, at higher applied voltage (15 V), after 1 h of treatment. At higher voltage applied (15V) to the laboratory electrodialysis system the value of energy consumption was higher (Wc for F3 was 147.33 kWh m-3 and Wc for C3 was 92.67 kWh m-3), possible due to the concentration polarization that becomes more prominent.

Synthesized polymeric membranes were characterized by FTIR-ATR, SEM and impedance measurements for proton conductivity. FTIR-ATR spectra showed the strong interactions of incorporation of SiO2 nanoparticles in polymer matrix with zinc ions. The main characteristic peaks of SiO2 were found at around 1078 cm−1 attributed to Si–O–Si and around 942 cm−1 ascribed to silanol (Si–OH) groups. SEM micrographs of the top-surface and cross-section of polymeric membranes with SiO2 nanoparticles showed that the SiO2 nanoparticles were uniform distributed and dispersed in the polymer matrix of the prepared polymeric membrane. The values of proton conductivity for all polymeric membranes increased with the increase of applied voltage. The impedance measurements for proton conductivity of polymeric membranes with SiO2 nanoparticles showed a considerable change. The higher value of proton conductivity for polymeric membrane with SiO2 nanoparticles was 0.840×103 S cm1 in comparison with the polymeric membrane without SiO2 nanoparticles that was 0.471×103 S cm1, after electrodialysis process, at 15 V.

The results suggest that the synthesized new polymeric membrane is excellent for efficient removal of metallic ions from wastewater and show promise for future applications (e.g., fuel cells, nanofiltration, electrodialysis reverse, ultrafiltration, reverse osmosis, microfiltration). Also, the new versatile laboratory electrodialysis system can future applied for water desalination, treatment of industrial effluents and removal/recover of other metallic ions, from different wastewaters, such as: Ni2+, Cu2+, Fe2+, Cd2+, Pb2+.

Point 2:  In the introduction several works are cited on membranes for pollutant removal that are not related to work. In particular, ceramic membranes. Instead, publications in which PVA membranes are used to remove arsenic or other contaminants are omitted. See for example:

Reversible swelling as a strategy in the development of smart membranes from electrospun polyvinyl alcohol nanofiber mats

Journal of Polymer Science  58(5), pp. 737-746 (2020)

Enhancing arsenic adsorption via excellent dispersion of iron oxide nanoparticles inside poly(vinyl alcohol) nanofibers

Journal of Environmental Chemical Engineering, Volume 9, Issue 1,February 2021, 104664

Biohybrid membranes for effective bacterial vehiculation and simultaneous removal of hexavalent chromium (CrVI) and phenol

Applied Microbiology and Biotechnology volume 105, pages827–838 (2021)

These works should be cited.

Response 2: Thank you for your suggestion.

In Introduction part with more information’s was improved and some section was removed and replaced with relevant information’s.

The recommended References was included for more clarity:

Cimadoro and Goyanes [47] prepared the poly(vinyl alcohol)-citric acid electrospun membranes for water treatment. The results indicated that the membrane achieves a rejection rate up to 99%. Pereira et al. [48] synthetized the biohybrid membranes by association of electrospun hydrolysis – resistant polyvinyl alcohol membranes and free-living bacteria for the removal of pollutants. The studies indicated that than 46% of the hexavalent chromium and the phenol content were removed from a tannery effluent. Torasso et al. [49] developed electrospun membranes containing iron oxide nanoparticles inside poly(vinyl alcohol) nanofibers for arsenic adsorption. They demonstrated that the prepared membrane has an enhanced arsenic adsorption capacity (52 mg/g).

Point 3:  The authors say: “the surface morphologies of the polymeric membrane (sample C0) were changed significantly with addition of SiO2 nanoparticles. However, is this due to the addition of SiO2, or is it due to the modifications produced by the process carried out to incorporate the nanoparticles? . Please explain.

Response 3: Thank you for your question.

More explanation was included for more clarity.

In order to study the effect of the incorporation of SiO2 nanoparticles in the structure of the prepared polymeric membranes, SEM micrographs at 2,000× magnification of the top surface and cross-section polymeric membranes without and with SiO2 nanoparticles (samples F0 and C0), before used in the laboratory electrodialysis system, were examined (Figure 6a-d).

Figure 6a-d showed the top-surface and cross-section morphologies of polymeric membrane without (F0) and with (C0) SiO2 nanoparticles. The top-surface view of the polymeric membrane without SiO2 nanoparticles (sample F0) (Figure 6a) showed the formation of spongy like holes, more and large pores (between 25.26 µm and 60.67 µm), nodule and macro void like structure [51]. The top-surface morphology of the polymeric membrane (sample C0) (Figure 6c) were changed significantly with incorporation of SiO2 nanoparticles in the polymer matrix. The SEM images, for sample C0, showed the uniform distribution and dispersion of the SiO2 nanoparticles in the active layer of the prepared polymeric membrane. It can be observed that the SiO2 nanoparticles were regular on surface of polymeric membrane. This indicated that the SiO2 nanoparticles were fixed on the polymer main chain. Also, SEM images indicate that the nanoparticles and polymer matrix were well compatible [59,63,66]. Moreover, the SiO2 nanoparticles incorporation in the prepared polymeric membrane, observed as white circular and non-circular shape, led to the formation of packed chain structure, reduce the pores size (between 1.045 µm and 288.5 nm) and reduce the macro void. The cross-section view of the polymeric membrane without SiO2 nanoparticles (Figure 6b) shows a fused nodule structure, whereas, the polymeric membrane with SiO2 nanoparticles had apparently loose nodule (Figure 6d). This can indicate the existence of interfacial stresses between SiO2 nanoparticles and polymer matrix. Figure 6d shows that the SiO2 nanoparticles are interconnected and densely incorporated and dispersed in the polymer matrix, which can be beneficial to improving the resistance and proton conductivity of the polymeric membrane [63-65].

Point 4:  The morphology of the composite material (C0) indicates that the material is anisotropic. What is the reason for the orientation that was generated, taking into account that the starting material is isotropic?

Response 4: Thank you for your question.

Anisotropic assembly of isotropic nanoparticles is observed in a polymeric membranes and can leads to  improvements in properties of the prepared polymeric membranes (rezistance, chemical stability, proton conductivity). The relatively simple sample preparation process means the approach could be used for large-scale manufacture of nanocomposites.

The cross-section view of the polymeric membrane without SiO2 nanoparticles (Figure 6b) shows a fused nodule structure, whereas, the polymeric membrane with SiO2 nanoparticles had apparently loose nodule (Figure 6d). This can indicate the existence of interfacial stresses between SiO2 nanoparticles and polymer matrix. Figure 6d shows that the SiO2 nanoparticles are interconnected and densely incorporated and dispersed in the polymer matrix, which can be beneficial to improving the resistance and proton conductivity of the polymeric membrane.

These membranes consist of several layers, each with different structures. A typical anisotropic membrane has a relatively dense, extremely thin surface layer supported on an open, much thicker porous substructure. The prepared polymeric membrane consists of a single membrane material, but the pore size change in different layers of the membrane.

Isotropic membranes have a uniform composition structure throughout, and they can be porous or dense. The resistance to mass transfer in these membranes are determined by the total membrane thickness. The simplest form of microporous membrane is a polymer film with cylindrical pores or capillaries. However, more commonly microporous membranes have a more open and random structure, with interconnected pores.

Point 5: What is the adhesion of the SI02 nanoparticles to the matrix? Adhesion tests should be carried out.

Response 5: Thank you for your question.

The aim of this study was to obtain polymeric membranes containing polyvinyl alcohol-SiO2 nanoparticles and to observe their effectiveness in the process of removing zinc ions through the electrodialysis process.

Also, the adhesion tests cannot be performed and included in the present manuscript, because is the subject of another manuscript that will be published in another Journal.

Point 6:  The authors show that indeed the membrane has the capacity to ad / absorb Zn2 +. What is the adsorption rate? How much is the adsorption capacity per gram of membrane?

Response 6: Thank you for your questions and suggestion.

The capacity to adsorption / absorption of Zn2 + cannot be included in the present manuscript, because is the subject of another manuscript that will be published in another Journal.

Round 2

Reviewer 1 Report

I’m still not satisfied with the current form of the manuscript. There is still room for improvement.

Abstract – Again, by replacing “novel” with “new” does not justify the PVA-SiO2 membrane is “new”. There are a lot of papers reporting PVA-SiO2 membrane before!

Besides, 65% removal of zinc is considered very LOW. There are many papers reported that membranes could achieve >90% or even complete removal of zinc.

Introduction is lengthy. The content should be concise and relevant to the TOPIC of study.

Line 63 - 100 ÷500 mg day-1. What does “÷” mean? Similar issue found in other pages as well.  

Line 69-78 – Bad grammar and sentence. There are in fact a lot of grammatical mistakes even though the authors have said revision is made!

Figure 2 and 3 should be combined as 1 image.

Equation (5) should be arranged in Methodology section

Table 3 should be presented in figure format.

Conclusion is too long. It should be brief but concise!!

There are 74 references cited for a simple technical paper. That is too much! Authors should reduce it to 40 references++. Cite only the most relevant ones!

Author Response

Response to Reviewer 1 Comments

I’m still not satisfied with the current form of the manuscript. There is still room for improvement.

Point 1. Abstract – Again, by replacing “novel” with “new” does not justify the PVA-SiO2 membrane is “new”. There are a lot of papers reporting PVA-SiO2 membrane before!

Response 1: Thank you for your remark.

Abstract was improved.

Point 2. Besides, 65% removal of zinc is considered very LOW. There are many papers reported that membranes could achieve >90% or even complete removal of zinc.

Response 2: Thank you for your remark.

The removal rate of zinc ions depends of many factors, such as: dimensions of the electrodialysis system, number of the compartments, dimensions of the membranes, number of the membranes, types of the membranes, composition of the membranes, composition of the wastewater, concentration of the zinc ions from feed wastewater, type of electrodes, operational conditions of the electrodialysis system (e.g., applied voltage, duration time for each test, temperature).

In the present manuscript are presented in details all operational conditions for electrodialysis system, to highlight why an extract percentage of over 65% was obtained. Comparative studies were also indicated, in order to be able to differentiate between the working conditions indicated in this manuscript and those in the literature.

The laboratory electrodialysis system was tested preliminary at different conditions (applied voltage at electrodes (3 V and 20 V)) and different concentration of wastewater containing zinc ions (between 0.5 g L-1 and 2.5 g L-1). The results showed that a lower applied voltage (3 V) and a low concentration of zinc ions (0.5 g L-1) leads to a low percentage removal of zinc ions. Moreover, a higher applied voltage (20 V) lead to increase of energy consumption. A concentrated solution (2.5 g L-1) leads to increased resistance of system, increased the energy consumption, destruction or damage to membranes, which go a long way to increasing the total costs. The electrodialysis system operated over 1 h, for treatment of wastewater containing zinc ions, led to a higher percentage removal of zinc ions, but the energy consumption was higher, and the prepared polymeric membranes showed cracks and finally decreases the membrane resistance. Due to these aspects mentioned above, which are not an advantage, in this study the laboratory electrodialysis system was chosen to be operated at values of applied voltages of 5 V, 10 V and 15 V, and a concentration of wastewater containing zinc ions of 2 g L-1.

Point 3. Introduction is lengthy. The content should be concise and relevant to the TOPIC of study.

Response 3: Thank you for your recommendation.

The Introduction part was improved. The most important and relevant information was kept.

Point 4. Line 63 - 100 ÷500 mg day-1. What does “÷” mean? Similar issue found in other pages as well.  

Response 4: Thank you for your suggestion.

÷ was replace with between

Point 5. Line 69-78 – Bad grammar and sentence. There are in fact a lot of grammatical mistakes even though the authors have said revision is made!

Response 5: Thank you for your suggestion.

The paragraph has been corrected and improved.

These conventional methods have limitations, such as: large amount of sludge, extra operational cost for sludge disposal, large amount of chemicals for removal of metals from waters and wastewaters, involves large activation energy, low selectivity [11-13]. Membrane technologies [14-16], such as: electrodialysis, ultrafiltration, reverse osmosis, nanofiltration, has been successfully applied in the last decades, at the laboratory scale and at the industrial level, due to the major advantages such as: high efficiency, can function at low temperature, does not require additional chemicals, low environmental impact [14,16-18].

Point 6. Figure 2 and 3 should be combined as 1 image.

Response 6: Thank you for your suggestion.

Figure 2 and 3 cannot be combined as 1 image because:

Electrodialysis ED is a separation process commercially used on laboratory level or a large scale for treatment of municipal or industrial effluents. ED system contains membranes and ions are transported through membranes from one solution to another under the influence of electrical potential difference used as a driving force. ED has been widely used in the desalination process and recovery of useful matters from effluents. The performance of ED depends on the operating conditions and device structures such as ion content of raw water, current density, flow rate, membrane properties, feed concentration, geometry of cell compartments.

In literature, electrodialysis (ED) is used to transport salt ions from one solution through ion-exchange membranes to another solution under the influence of an applied electric potential difference. This is done in a configuration called an electrodialysis cell. The cell consists of a feed (dilute) compartment and a concentrate (brine) compartment formed by an anion exchange membrane and a cation exchange membrane placed between two electrodes. In almost all practical electrodialysis processes, multiple electrodialysis cells are arranged into a configuration called an electrodialysis stack, with alternating anion and cation-exchange membranes forming the multiple electrodialysis cells. Electrodialysis processes are different from distillation techniques and other membrane based processes (such as reverse osmosis (RO)) in that dissolved species are moved away from the feed stream rather than the reverse. Because the quantity of dissolved species in the feed stream is far less than that of the fluid, electrodialysis offers the practical advantage of much higher feed recovery in many applications.

The impedance spectroscopy represents one of the essential techniques that allow the study of the electrical response of the membranes in terms of complex impedance, dielectric constant, or conductivity.

Electrochemical impedance is the response of an electrochemical system (cell) to an applied potential. The frequency dependence of this impedance can reveal underlying chemical processes.

Theory. EIS is a powerful characterization technique that can provide detailed information about the electrode-electrolyte interface of an electrochemical system. Potentiostatic EIS is the most common variant of this technique, where a small amplitude sinusoidal potential signal (e) at a given frequency is superimposed onto a constant potential (Ec).

The alternating current (ac) response (i) of the electrochemical system is usually out of phase with the potential signal due to the existence of non-faradaic processes (e.g., double layer capacitance) and reaction of adsorbed species, which act as capacitors and inductors, respectively

Point 7. Equation (5) should be arranged in Methodology section

Response 7: Thank you for your suggestion.

Equation (5) was included in Methodology section.

Point 8. Table 3 should be presented in figure format.

Response 8: Thank you for your suggestion.

Table 3 was deleted and was presented as Figure.

Point 9. Conclusion is too long. It should be brief but concise!!

Response 9: Thank you for your remarks and suggestion.

The Conclusions section has been shortened and contains only the relevant information obtained from the studies performed.

In this paper, the polymeric membranes without and with SiO2 nanoparticles were successfully prepared and applied for the removal of zinc ions (Zn2+) from a simulated wastewater using a new versatile laboratory electrodialysis system. The experimental results from the electrodialysis process indicated that the values of demineralization rate and of percentage removal of zinc ions increase with the increasing of applied voltage. This can be due to migration of water that solvated the zinc ions. The results revealed that the higher value of percentage removal of zinc ions (66.58%) was obtained for the polymeric membrane with SiO2 nanoparticles, at higher applied voltage (15 V), after 1 h of treatment. FTIR-ATR spectra showed the strong interactions of SiO2 nanoparticles incorporated in polymer matrix with zinc ions. SEM micrographs of the top-surface and cross-section of polymeric membrane with SiO2 nanoparticles indicated that the SiO2 nanoparticles were uniform distributed and dispersed in the polymer matrix of the prepared polymeric membrane. The values of proton conductivity for all polymeric membranes increase with the increasing of applied voltage. The impedance measurements for proton conductivity of polymeric membranes with SiO2 nanoparticles suggested a considerable change.

The results demonstrated that the synthesized polymeric membrane is excellent for efficient removal of metallic ions from wastewater and can be used for future applications (e.g., fuel cells, nanofiltration, electrodialysis reverse, ultrafiltration, reverse osmosis, microfiltration). Also, the new versatile laboratory electrodialysis system can future applied for water desalination, treatment of industrial effluents and removal/recover of other metallic ions (e.g., Ni2+, Cu2+, Fe2+, Cd2+, Pb2+) from different wastewaters.

Point 10. There are 74 references cited for a simple technical paper. That is too much! Authors should reduce it to 40 references++. Cite only the most relevant ones!

Response 10: Thank you for your remarks and suggestion.

References was reduced. In the manuscript were included only relevant References.

Reviewer 3 Report

The authors have satisfactorily answered most of my questions. The manuscript in its current state deserves to be published

Author Response

Thank you for the positive comments.

Round 3

Reviewer 1 Report

Authors have revised the manuscript according to the comments.